# Efficiency of "Publish or Perish" Policy—Some Considerations Based on the Uzbekistan Experience

**Bahtiyor Eshchanov** [1,*], **Kobilbek Abduraimov** [2], **Mavluda Ibragimova** [3] and **Ruzumboy Eshchanov** [4]

1 Center for Economic Research and Reforms, Tashkent 100043, Uzbekistan
2 Economics Department, Westminster International University in Tashkent, Tashkent 100047, Uzbekistan; kabduraimov@outlook.com
3 Institute of General and Inorganic Chemistry, Academy of Sciences, Tashkent 100052, Uzbekistan; mavluda@gmail.com
4 Chirchiq State Pedagogical Institute in Tashkent Region, Chirchiq 111700, Uzbekistan; ruzimboy@gmail.com
* Correspondence: b.eshchanov@gmail.com; Tel.: +998-914330039 or +998-971420888

**Abstract:** Researchers from Uzbekistan are leading the global list of publications in predatory journals. The current paper reviews the principles of implementation of the "publish or perish policy" in Uzbekistan with an overarching aim of detecting the factors that are pushing more and more scholars to publish the results of their studies in predatory journals. Scientific publications have historically been a cornerstone in the development of science. For the past five decades, the quantity of publications has become a common indicator for determining academic capacity. Governments and institutions are increasingly employing this indicator as an important criterion for promotion and recruitment; simultaneously, researchers are being awarded Ph.D. and D.Sc. degrees for the number of articles they publish in scholarly journals. Many talented academics have had a pay rise or promotion declined due to a short or nonexistent bibliography, which leads to significant pressure on academics to publish. The "publish or perish" principle has become a trend in academia and the key performance indicator for habilitation in Uzbekistan. The present study makes a case for re-examining the criteria set by the Supreme Attestation Commission of the Republic of Uzbekistan for candidates applying for Ph.D. and D.Sc. as well as faculty promotion requirements in the light of current evidence for the deteriorating academic performance of scholars.

**Keywords:** publish-or-perish policy; academia; predatory publishing; open-access publishing; Beall's criteria; binary regression model; Uzbekistan

## 1. Introduction

In academia, the phrase "publish or perish" is more than a pervasive culture; it reflects a harsh reality—one in which scholars are under immense pressure to publish their research findings in scholarly journals in order to advance their careers. Scientific publications are usually considered a significant metric for academic performance. This pressure, in turn, provides a major incentive for academics to submit their manuscripts to journals.

As a result, a great deal of attention has recently been devoted to whether the publication of scholarly articles should be the universal criteria for advancement in the academic community, given the relative ease with which academic papers can be published in low-quality journals. The implementation of this policy—which is referred to as publish or perish—produces, among contemporary scholars, an immediate and sometimes surprising set of responses—ranging from enthusiasm to anger. Supporters view this as an opportunity for career advancement, whereas opponents express their disagreement by citing the deteriorating quality of research papers. The depth of feelings on both sides often leaves policymakers in a bind, wondering exactly what policies lead to a meaningful compromise between the two sides.

Authorities in Uzbekistan have also adopted this policy, intending to increase the local researchers' scientific outreach and recognition of the achievements and results. However, this has yielded opposite results by drastically increasing the number of publications in predatory journals by Uzbekistani scholars. For the third year in a row, researchers from Uzbekistan lead the list of countries whose scholars are publishing in predatory journals (Table 1).

**Table 1.** Percentage of articles and reviews by countries in sources discontinued by Scopus in 2020.

|    |            | Total Articles and Reviews | In Discontinued Journals | % in Discontinued Journals |
|----|------------|-----------|---------|---------|
| 1  | Uzbekistan | 2916 | 1740 | 59.67% |
| 2  | Iraq | 14,687 | 3643 | 24.80% |
| 3  | Indonesia | 25,601 | 5995 | 23.42% |
| 4  | Philippines | 4497 | 550 | 12.23% |
| 5  | Kazakhstan | 4070 | 444 | 10.91% |
| 6  | Malaysia | 28,527 | 2833 | 9.93% |
| 7  | India | 155,294 | 13,383 | 8.62% |
| 8  | Jordan | 6096 | 389 | 6.38% |
| 9  | Bahrain | 896 | 57 | 6.36% |
| 10 | Ukraine | 12,521 | 700 | 5.59% |
| 11 | Kyrgyzstan | 413 | 20 | 4.84% |
| 12 | Afghanistan | 268 | 11 | 4.10% |
| 13 | Vietnam | 15,451 | 576 | 3.73% |
| 14 | Romania | 11,984 | 443 | 3.70% |
| 15 | Morocco | 7196 | 260 | 3.61% |
| 16 | Oman | 2278 | 81 | 3.56% |
| 17 | Papua New Guinea | 217 | 6 | 2.76% |
| 18 | Peru | 4288 | 111 | 2.59% |
| 19 | Palestine | 1087 | 28 | 2.58% |
| 20 | Tajikistan | 236 | 6 | 2.54% |

Source: Kennesov. 2020 [1].

Our research was undertaken in response to growing evidence that the academic environment in Uzbekistan had undergone radical changes. In particular, we discuss an emerging trend toward "predatory publishing"—research publications that publish articles without providing peer-review services. To date, there has been comparatively little modeling on factors that drive the likelihood of the journals being predatory despite recognizing the extent of the problem. In this paper, we present a pioneering model to explain common characteristics among predatory journals. We also show that scholars' academic performance has deteriorated over the past decade in response to the publish-or-perish policy.

The present study provides an important opportunity for the Supreme Attestation Commission of the Republic of Uzbekistan to consider how the academic performance of scholars has changed in response to the practically impossible publication requirements for researchers, applicants for Philosophy Doctor (Ph.D.) and Doctor of Science (D.Sc.) degrees. It also develops recommendation for creating a research-enabling environment at institutions and universities of Uzbekistan.

## 2. Subscription-Based Journals against Open-Access Movement

Scientific research plays a major role in the evolution of science, which, in turn, leads to human progress by helping nations to address global challenges—from economic to environmental. The rate at which scientific research progresses inevitably depends on international collaboration and exchanging research findings.

Public access to research findings is now such a part of everyday life that it is often taken for granted by the public; however, this was not always the case, nor did it appear inevitable or predictable. The future of scientific publications changed forever in 2001

when the Open Society Foundations called for a radical change in the research sphere by creating new open-access journals and urging researchers to make the results of their work publicly available [2]. This transition pioneered an open-access movement (OA)—a model for publishing scholarly, peer-reviewed journals online free of charge. The advent of the Internet played a crucial role in bringing this vision to reality.

Subscription-based publishing has been the conventional model until recently [3,4]. The subscription-based model creates barriers to information and knowledge sharing, creating significant difficulties for research and education fields. Subscription fees to scholarly journals increase annually by 8–10%, outpacing the inflation rate measured by the consumer price index (CPI) [5]. Articles and publications behind the paywall are not accessible to 80–90% of the research community, especially in developing countries. Researchers could not conduct research because they did not have access to the contemporary trends and dynamics in their field because of the paywall. To avoid huge subscription fees, big libraries including Harvard, North Carolina, Cornell, and MIT launched initiatives to switch off subscriptions [6–8], partnering with scientific societies to diminish the cost related to subscription-based publications and enhancing research and knowledge-based communication. The open-access model has been seen as a potential solution to overcome barriers of subscriptions.

Open-access journal publishing is rapidly growing in popularity and has already succeeded in strengthening its position as a rich source of journal articles and research findings. Open-access journals have significantly changed the financial model of journal publishing. Historically, journals relied financially on subscription fees. In contrast, most OA publishers use business models where academics are charged for their publications. Of course, numerous OA publishers are funded by dedicated agencies and funds, which do not require an APF for publishing.

The potential for revenue from author fees and the relative ease with which research papers are published online led to the emergence of so-called predatory publishers—those that publish misleading journals to imitate open-access models, more often than not, without providing editorial and publishing services. These journals not only undermine the OA movement but also pose a serious threat to the research integrity.

The history of the open-access movement begins with the concern of librarians toward the ever-increasing subscription fees of scholarly journals, which was soon reinforced by the advent of the Internet [9]. The subscription prices increased for several reasons. First, as the baby boomer generation reached the age where many were finishing their Ph.D. degrees, scholarly journals began to publish more articles to accommodate the ever-increasing number of research papers [10]. Another key factor contributed to an increase in subscription fees—the advent of new fields of study including but not limited to nanotechnology and artificial intelligence—a phenomenon that occurred parallel to when baby boomers began to occupy higher education faculty positions [10]. Commercialization of scientific publishing has also played an important role in increasing subscription fees.

To meet the supply for the number of research articles being submitted for publication, many journals had to change the frequency with which journals were published, from bi-yearly to quarterly, from quarterly to monthly. Consequently, a small number of large commercial publishers who had acquired most of the top journals in many fields by the early 1990s began to adopt a variety of profit-oriented pricing strategies such as bundling together a large group of journals to which libraries were required to subscribe to obtain access to the key journals [11]. Due to these practices—that many see as unethical—many institutions could not afford the subscriptions for the most important research being conducted in the STEM fields (science, technology, engineering, and mathematics) [11]; in turn, the term "serials crisis" emerged.

The serials crisis, along with the emergence of the Internet in the mid-1990s, led to the open-access movement. The foundation for open access was laid by Paul Ginsparg in 1991, who established the arXiv repository at the Los Alamos National Laboratory to make preprints in physics freely accessible [12]; however, the international collaborative effort

accelerated even further after three ambitious proclamations were made in the early 2000s following scholarly meetings in Germany, the USA, and Hungary and subsequently turned into more of an institution rather than a social movement. Finally, the statements came to serve as a substitute for thought [13].

The state-sponsored organizations, such as the National Institutes of Health, also supported this initiative by requiring the research they fund to be publicly available, which further broadened and strengthened the base of support for the open-access movement [14]. This evolution was strongly supported worldwide for several reasons. First, OA articles reach broader audiences than print-based publications. Second, unlike traditional publishing, research materials in the OA model are not restricted to articles.

Any digital content, such as videos, raw and processed data, and software, can be included in a digital archive, which takes the research field to a completely new level. Third, access to knowledge is not limited by the budget available to a library irrespective of its geographical location. The digitalization of journals also allows scholars to search entire runs of publications with research from as far back as the 19th century, which saves time and helps avoid the duplication of works.

Soon after, several commercial organizations such as BioMed Central and the Public Library of Science entered the market by offering a different financial model for the funding of research—authors are charged for their publications, ranging from USD 500 in 2002 to USD 2500 in 2006, because providing researchers with high-quality peer-reviewed scientific material free of charge is very difficult due to the high costs incurred by the publisher [14].

The emergence of predatory journals has undermined the open-access environment. Jeffrey Beall, a librarian from the University of Colorado, first coined the term "predatory publisher" in 2010. Predatory publishers exploit open-access publishing and charge an article-processing fee (APF) by promising, but not providing, publishing services in return; they do not meet ethical scholarly publishing industry standards and seek only profit from the APF.

It must be emphasized that some research communities and authorities were concerned about quality assurance in OA publishing while maintaining affordability to a broader community of scholars. These communities started funding the OA publishers, paving the way to so-called "diamond" or "platinum" OA publishing, under which articles are published without charging an APF. The number of these diamond and platinum OA journals is increasing.

Since predatory journals aim to generate maximum profit by accepting all the papers submitted, open-access journals have a strong conflict of interest regarding peer review. This conflict is at the heart of the ongoing downfall of scholarly publishing [15]. Hence, predatory journals are designed to publish manuscripts rejected by peer-reviewed publishers such as Springer, Elsevier, Wiley, Taylor & Francis, Oxford University Press, MDPI, and Frontiers [15].

John Bohannon highlighted the issue [16] by simultaneously submitting an unprofessional article to 304 different open-access journals. According to him, the article has so many "grave errors that a competent peer reviewer should easily identify it as flawed and unpublishable"; however, more than half of the publishers accepted the manuscript.

According to the findings of Beall, the number of predatory publishers increased from 18 in 2011 to 693 in 2015 (cited in Ayeni and Adetoro [17], p. 18). John Bohannon then conducted extensive research on predatory journals by compiling a list of publishers who unprofessionally exploit the OA model for profit, also known as Beall's List of Predatory Publishers. After that, he and other authors designed a set of criteria for identifying predatory journals, some of which include aggressive marketing through emails, the imitation of top journals' names, little geographical diversity among the editorial board and authors, and quick acceptance of articles.

## 3. Publish-or-Perish Policy

The frequency with which scholars produce publications is one of the main metrics to demonstrate academic talent—helping to climb the career ladder, protect their jobs, and secure funding for their institutions. Globally, academic institutions use the number of publications to measure competency, with many universities increasingly using this as the main criterion during recruitment. Scholars who publish infrequently or whose focus is directed toward other agendas, such as instructing undergraduates, may eventually find themselves out of competition. The publish-or-perish mentality and subsequent threat to academic employment create immense pressure to publish. The term "publish or perish", initially coined by Coolidge [18], is now becoming a harsh reality.

Publish or perish is defined as the pressure in academic discipline to frequently publish research papers in scholarly journals to sustain and advance one's career. Academics who fail to publish eventually perish by either not finding jobs or losing their existing positions. Perish may also refer to the denial of a promotion. Whether the work they are publishing makes a measurable impact on their field of study is, unfortunately, a secondary concern.

The publish-or-perish policy increased in popularity in the higher education institutions of developing economies where Ph.D. and D.Sc. degrees became the main criteria for faculty positions at universities and research institutions [19]. The policy dictates that academics report publications to meet "expectations" and "objectives" in the work plan [19]. Of course, academic and research departments aim for a list of publications in top journals; however, publishing in top journals is a difficult and time-consuming process; scholars and academics face a trade-off: publishing in predatory journals or "perishing".

Predatory publishers have become an urgent issue for scientific research as they strive to generate income from authors without offering proper peer-review services [20,21]. When young researchers are put under pressure to publish their work in international journals to obtain a degree or promotion, predatory journals flourish. Predatory journal publishers regularly engage in advertising campaigns that market their rapid peer-review processes and accelerated publication services in days, even in hours, and rarely require revisions [22,23].

Nevertheless, socio-political systems in emerging economies could not sustain open-access journals of international renown [19]. The primary requirement was to publish a paper in internationally recognized journals. Due to lagging behind in terms of modern education, emerging economies had few journals that satisfied the requirements for international recognition; however, it was not easy for academics to have their work published in international journals, mainly due to the poor quality of the studies and cost of publication [19]. These factors led academics to have their work published in predatory journals.

The publish-or-perish policy and academic performance of the scholars in the former Soviet Union and Eastern Bloc member countries have been investigated in numerous studies. For instance, Kozak et al. investigated the scientific performance and international scientific cooperation processes in former Warsaw Pact member countries to reveal the impact of the collapse of the Soviet Union on research publication [24]. Grančay et al. investigated the performance of economists from Central and Eastern European countries under ever-increasing publication requirements [25]. Neither study covered the performance of scholars from Uzbekistan due to a lack of data.

Despite a lack of literature investigating Uzbekistan, the condition of scholars from Uzbekistan was no different. Numerous policies implemented in the scientific sphere did not increase the country's scientific reputation but instead harmed it.

The most recent decision of the authorities envisages offering the D.Sc. diploma of a national standard to Ph.D. degree holders from top-1000 universities based on the Times Higher Education, QS, and the Shanghai World University Rankings. This policy does not take the candidates' previous or current scientific performance into account, which again leads to an increasing number of pseudo-scholars alongside decent scholars.

The Supreme Attestation Commission under the Cabinet of Ministers of Uzbekistan introduced the requirement for completion of Ph.D. and D.Sc. degrees as follows:

> *"Prior to defending a doctoral dissertation (D.Sc.), applicants must:*
>
> - *provide a broad discussion of the results of the dissertation at international and national scientific and scientific-practical conferences;*
> - *have at least ten scientific articles (including one international), reflecting the main results of the dissertation, published in scientific journals determined by the Higher Attestation Commission.*
>
> *Applicants for the degree of Doctor of Science (D.Sc.) in Social Sciences and Humanities must publish a monograph based on the results of the dissertation."*[1]

The resolution above delegates to the Supreme Attestation Commission the power to create a list of scientific journals accepted for degree completion. The Supreme Attestation Commission regularly updates this list of journals. Almost all the predatory journals in which scholars from Uzbekistan have published their studies are included in that list, enabling the candidates to defend their degrees successfully.

Another driver for increased predatory publications among scholars from Uzbekistan is the extremely high requirements introduced for the attestation of the scholars working in scientific and research institutions. Table 2 describes the attestation requirements for the staff of research institutions under the Academy of Sciences, effective for 2021:

**Table 2.** Category of attestation requirements.

| | |
|---|---|
| A | Scientific articles published in 2018–2020 in peer-reviewed journals, indexed in the Scopus (scopus.com) and Web of Science (webofknowledge.com, publons.com) databases |
| B | Intellectual property protection documents—patents and applications indexed in the PATENTSCOPE (wipo.int/patentscope/en/) and Intellektual Mulk (newbaza.ima.uz/, baza.ima.uz/) databases during 2018–2020 |
| C | Specifications and technical documentation approved by the competent authorities, justifying the implementation of the invention/discovery to practice during 2018–2020 |
| D | Sale of products and services for at least UZS 100 million per member of staff being attested during 2018–2020, confirmed by relevant documents |

- *For chief researchers and heads of laboratories and departments, the number of items to be presented must be at least two items from Category A and B of Table 1, e.g., two articles from the databases or one article from the below databases and one patent or two patents.*
- *For leading and senior researchers, any two items from category A, B, C, or D of Table 2.*
- *For junior researchers, it is mandatory to have two articles published in scientific journals during 2018–2020.*

Moreover, authorities overseeing the scientific performance of scholars and researchers in the country, namely, the Supreme Attestation Commission, the Ministry of Innovative Development, and the Academy of Sciences, launched and operated various web platforms, which, among other functions, are used for the monitoring and evaluation of the scientific performance of the scholars. For example, www.salohiyat.uz and www.fanportal.mininnovation.uz are used to rank scholars. The problem with these platforms is that authors are required to enter their data manually, and authorities cannot evaluate the trustworthiness and credibility of the entries. The above portals are not integrated into third-party databases such as Scopus Authors database, SSRN, ORCID, ResearchGate, Google Scholar, and RePEc/Ideas platforms, which would allow for the maintaining of a decent level of objectivity and fairness in evaluating the research performance of the scholars.

## 4. Methodology

### 4.1. Data Description

The present study draws its data from the primary source of information on the public educational network in Uzbekistan: ZiyoNET[2]. The data cover a representative sample of

2025 individuals who obtained either a Ph.D. or D.Sc. degree from January 2016 to March 2019. Given our interest in evaluating the academic performance of scholars based on the quality of the journal in which their articles were published, only international journals were included. The exclusion of individuals whose articles have not been either published in English or international journals brings the sample to 1110 scholars. Figure 1 illustrates the annual percentage of academics who published their articles according to the criteria. In general, over the past four years, two out of three academics in Uzbekistan published their work in English and international journals.

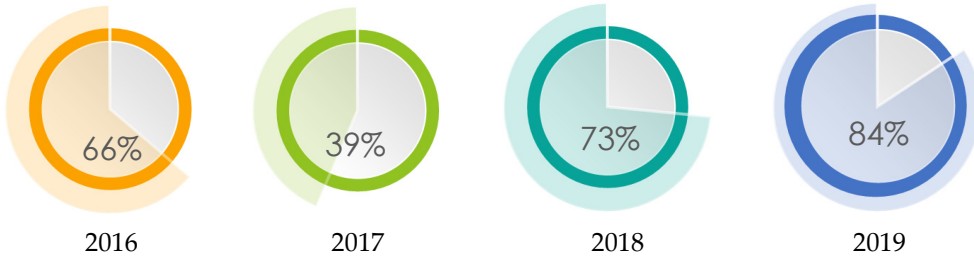

**Figure 1.** Percentage of scholars whose articles were published in English and international journals.

It is important to note that, between 2011 and 2017, Uzbekistan practiced a single-stage habilitation policy according to which any graduate-level degree holders (B.Sc., M.Sc., and Ph.D.) could equally pursue a doctoral degree program and be awarded the highest D.Sc. degree. From 2017, however, a D.Sc. is awarded for a portfolio of work later in an academic's career, that is, after a Ph.D. Figure 2 illustrates the percent of academics who obtained a D.Sc. compared to a Ph.D. between 2016 and 2019. The number of D.Sc. degrees progressively decreased from 100% to 32% over the last four years.

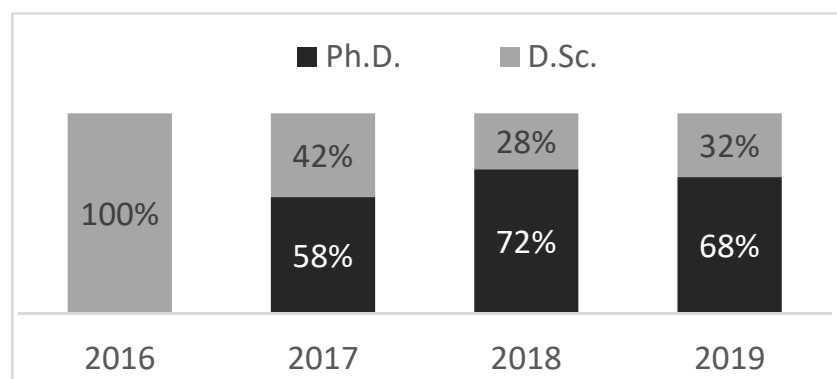

**Figure 2.** Percentage of academics who obtained D.Sc. compared to Ph.D.

Furthermore, the data provide an extensive set of characteristics for each individual, such as academic discipline, the level of university degree attained, journal title, and year of publication. In addition, each set of characteristics for each individual includes a reference to the institution or university for which the author is working while obtaining the degree. By assembling these different pieces of information, a matched panel data set was built, corresponding to nearly 2500 articles published in 610 international journals by 1110 authors from 125 institutions in 30 academic fields across Uzbekistan. As Figure 3 demonstrates, the top subjects are medicine, physics, technical science, and chemistry, which collectively account for 50% of the total.

| | | Articles | Percentage | PhD | DSc |
|---|---|---|---|---|---|
| ⊕ | Medicine | 545 | 21.78% | 30% | 70% |
| ⚛ | Physics | 322 | 12.87% | 32% | 68% |
| ⚙ | Technical Science | 258 | 10.31% | 47% | 53% |
| ⚗ | Chemistry | 203 | 8.11% | 40% | 60% |
| 📖 | Philology | 138 | 5.52% | 70% | 30% |

**Figure 3.** Top-five academic disciplines based on the number of publications 2016–2019.

Figure 4 compares universities and institutions based on the number of international publications. With almost 220 publications, the National University of Uzbekistan ranks first among 125 institutions, followed by Tashkent Medical Academy and Tashkent State Technical University; each published approximately 125 articles in international journals.

| | | Articles | Percentage | PhD | DSc |
|---|---|---|---|---|---|
| 1 | National University of Uzbekistan | 219 | 8.75% | 31.05% | 68.95% |
| 2 | Tashkent Medical Academy | 126 | 5.04% | 19.05% | 80.95% |
| 3 | Tashkent Technical University | 122 | 4.88% | 56.56% | 43.44% |
| 4 | Samarkand State University | 96 | 3.84% | 33.33% | 66.67% |
| 5 | Tashkent University of IT | 85 | 3.40% | 61.18% | 38.82% |
| 6 | Center of Endocrinology | 73 | 2.92% | 64.38% | 35.62% |
| 7 | Institute of Mathematics | 64 | 2.56% | 21.88% | 78.13% |
| 8 | Tashkent Pedagogical University | 60 | 2.40% | 70.00% | 30.00% |
| 9 | Institute of Chemistry | 58 | 2.32% | 37.93% | 62.07% |
| 10 | Tashkent University of Economics | 57 | 2.28% | 57.89% | 42.11% |
| | Others Institutions | 1,542 | 61.63% | 34.93% | 65.07% |

**Figure 4.** Top-ten universities and institutions based on the number of publications.

Finally, the descriptive statistics of the publication data set for 2016 and 2019 are as follows: the average number of articles per author is 2 publications, whereas the most frequent number of publications per author is one paper with the standard deviation of 2 publications; the maximum number of publications is capped at 19 publications. Further categorized information is provided in the next chart as a histogram (Figure 5).

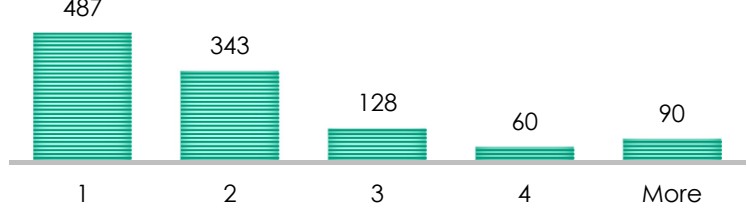

**Figure 5.** Histogram of number of international articles per author.

Overall, out of 1110 scholars, almost 45% of the authors published only one article in international journals, whereas one-third published two articles, and the maximum number of articles published by a scholar was 19.

### 4.2. Estimation Techniques

In order to present an accurate picture of the current academic environment, the study approaches the analysis from two perspectives. The first approach deals with qualitative analysis by employing Beall's criteria, which identified several characteristics to detect



a predatory journal. These criteria are classified under four broad categories—editorial criteria, submission and peer-review process, policies and publication fees, and website design—all of which are presented in the table below.

Researchers have previously investigated the predatory publishing problem for Kazakhstan in 2018, such as Bulat Kenesov. He employed the Scopus scientific citation database to determine the rate of predatory publishing [26]. He reported that the Scopus database detected and excluded the predatory journals from indexing in 2018, in which the largest number of authors corresponded to scientists from his country. In contrast to the situation in Kazakhstan, not all articles published by Uzbek scientists are included in this database, and hence, Beall's criteria to identify predatory publishing rate is used based on the data collected from ZiyoNet.

Table 3 demonstrates the classification based on Beall's criteria. Publishers are classified as either predatory or credible, according to this table. The focus of this method is primarily directed to identifying academic disciplines and universities involved in predatory publishing practices in Uzbekistan.

**Table 3.** Beall's criteria for classifying predatory journals.

| Criteria Group | Criteria | Metric | Weight |
|---|---|---|---|
| Editorial section | Email and contact of editor by department | Official email<br>General email services | 0<br>1 |
| | Number of editors | More than 6–10 per subject (based on scope)<br>Less than 5 | 0<br>1 |
| | Diversity of the board | Geographical diversity among editors<br>Little geographical diversity among editors | 0<br>1 |
| Submission and review process | Submission system | Professional manuscript submission system<br>By email or direct uploading | 0<br>1 |
| | Peer-review duration | More than one month (up to two years)<br>Less than a month | 0<br>1 |
| | Supplementary materials | Allows and instructs authors to share their data<br>Does not facilitate sharing electronic files | 0<br>1 |
| Ethics and publication fees | Publication fee | Relatively high (from USD 1000 to USD 3000)<br>Very low (up to USD 500) | 0<br>1 |
| | Publication ethics | Member of Committee on Publication Ethics (COPE)<br>Not member of COPE | 0<br>1 |
| Website design and scope of the journal | Advertisement | No advertisement<br>Spam emails | 0<br>1 |
| | Scope of the journal | Journal's scope is very specific<br>Covers broad fields of science | 0<br>1 |
| | Indexes and metrics | Indexed by Web of Science, PubMed, Scopus<br>Fake indexes in important databases | 0<br>1 |

### 4.3. The Model

In the second part, we construct a model to explain a set of key determinants that increase the likelihood of a journal being predatory. Since a journal is either predatory or credible, the dependent variable itself is qualitative, i.e., a binary regression model. The model constructed for the analysis is as follows:

$$Y_{it} = \alpha + \sum_{j=1}^{J} \beta_j X_{j,it} + \sum_{k=1}^{K} \delta_k Z_{k,it} + \varepsilon_{it} \tag{1}$$

where $X$ is a vector of the individual characteristics of academics such as a dummy variable relating to a degree of academic achievement, which is 1 if a scholar is a D.Sc. and 0 if has

a Ph.D. Z contains a set of journal's characteristics in which an article is published, such as whether the journal claims to be international (title includes: "international", "global", "world", "European", and "American") or contains such general keywords as "scientific", "research", "advanced", or "review", which makes the scope of the journal very broad; $\alpha$ is the constant; $\beta$ and $\delta$ are the parameters to be estimated.

The present study employs three approaches to developing a probability model for binary regression: the linear probability model (LPM) and logit and probit models. Since the linear probability model estimates the conditional probability of the event Y occurring given X, it should lie between 0 and 1. Although this should hold true, LPM does not guarantee that its estimators will fulfill this restriction; however, the logit and probit models guarantee that the estimated probabilities will indeed lie between the logical limits 0 and 1.

It should also be noted that, when a regression model involves time series data, there is a high probability that there is a structural change in the relationship between output and input variables [27]. A structural change refers to a circumstance in which the values of the model's parameters may not remain the same throughout the entire period [27]. Policy changes could cause this. If this is ignored, the regression model assumes that the link between dependent and independent variables has not changed much over 23 years, and this is a very unrealistic assumption.

Hence, we introduce a dummy variable for the periods, which is 1 if publications date from 1996 to 2010 and 0 otherwise, to test this hypothesis. The choice of the time period is justified because, between 2008 and 2010, the financial model of the publishing industry shifted from readers to authors; therefore, a priori, it is expected that the predatory rate is higher for the periods after 2010.

## 5. Results and Discussion

### 5.1. Predatory Publishing Trend

Figure 6 below illustrates the percentage of articles published in predatory journals. The graphical representation clearly shows that until 2010, nine out of ten Uzbek scholars published their research findings in credible international journals; however, by 2018, the percentage of academics who publish their research output in influential journals progressively decreased to only 2%. In other words, currently, 98 articles out of 100 are published in predatory journals.

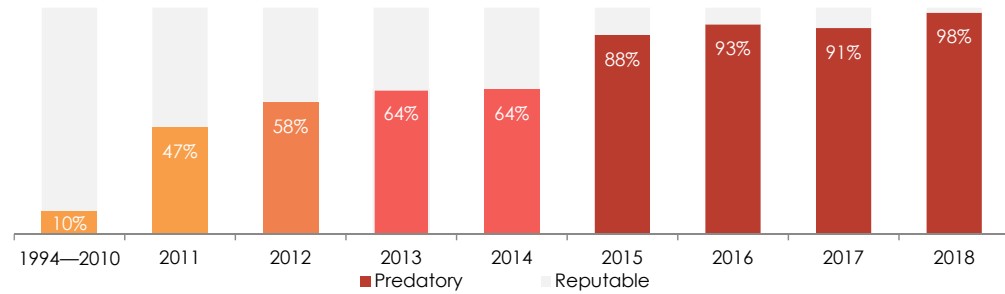

**Figure 6.** Histogram of number of international articles per author.

According to the most recent analysis presented by Bulat Kenessov, scholars from Uzbekistan are still world leaders in predatory publishing, having published 2916 articles in 2020, 1740 of which were in journals that were excluded from indexing in the Scopus database. The number would be much higher if a greater number of the journals from the Supreme Attestation Commission's recommended journals list is not indexed at all.

### 5.2. Qualitative Analysis by Journals

Figure 7 below presents the top-five reputable journals in which Uzbek scholars have published their articles. Interestingly, Springer—a reputable global publishing company—

issues two online journals from Uzbekistan: *Chemistry of Natural Compounds* (CNC) and *Applied Solar Energy* (ASE).

| | | Publications | Subject | Publisher | Country |
|---|---|---|---|---|---|
| | Chemistry of Natural Compounds | 45 | Chemistry | Springer | Uzbekistan |
| | Applied Solar Energy | 40 | Physics | Springer | Uzbekistan |
| | Physical Review | 31 | Physics | American Physical Society | USA |
| | Semiconductors | 14 | Physics | Springer | Russia |
| | Ukrainian Journal of Physics | 13 | Physics | Naukova Dumka | Ukraine |

**Figure 7.** Top-five scholarly journals based on the number of articles.

These two journals are the only internationally recognized journals from Uzbekistan listed in Springer. Although these journals are originally published in Russian and are translated into English in Russia and the Netherlands, where the Publishing Editorial Offices are located, the Editorial Boards are headquartered in Uzbekistan. The editorial board of CNC is geographically diverse and employs 28 reviewers from China, Russia, Kazakhstan, Cyprus, Georgia, Ukraine, and Pakistan. Conversely, all 12 editors of ASE come from Uzbekistan. This clearly shows the extent to which editorial boards from Uzbekistan in chemistry and physics contribute to the peer-review process for scientific articles in research.

Figure 8 presents a list of top-five predatory journals in which academics from Uzbekistan publish their research outputs, none of which is indexed in official citation databases. *European Science Review* ranks first in the list of predatory journals in which 20% of the predatory research papers were published, followed by *European Applied Sciences* (5%) and *Austrian Journal of Technical and Natural Sciences* (4%).

| | | Publications | Subject | Publisher | Country |
|---|---|---|---|---|---|
| | European Science Review | 368 | Multidisciplinary | Premier Publisher | Austria |
| | European Applied Sciences | 101 | Multidisciplinary | ORT Publishing | Germany |
| | Austrian Journal of Technical and Natural Sciences | 75 | Multidisciplinary | Premier Publisher | Austria |
| | Eastern European Scientific Journal | 69 | Multidisciplinary | Auris | Germany |
| | Medical and Health Science Journal | 54 | Medicine | PRADEC | Czech Republic |

**Figure 8.** Top-five predatory journals based on the number of articles.

As mentioned above, the Supreme Attestation Commission of Uzbekistan compiles a list of recommended journals in which to publish the basic scientific results of dissertations. The first four journals were included in the list in 2015; however, the subsequent edition (published in 2016) excluded two journals from the list: *European Applied Sciences* and *Eastern European Scientific Journal*. From 2012 and 2016, by publishing articles in these journals, 121 academics obtained Ph.D. (42%) and D.Sc. (58%) degrees. Despite being excluded from the list, 30 authors still obtained a Ph.D. (70%) and D.Sc. (30%) by publishing articles in those journals between 2017 and 2018; however, the latest edition of the list of recommended journals published in 2018 still contains several predatory journals, including *European Science Review* and *Austrian Journal of Technical and Natural Sciences* (Supreme Attestation Commission, 2018).

For illustrative purposes, two types of publishers were chosen—Springer and Premier Publishing (which publishes *European Science Review* and *Austrian Journal of Technical*

*and Natural Sciences*)—to distinguish predatory journals from credible journals based on Beall's criteria.

### 5.3. Editorial Members Criteria

First, most predatory journals rarely register official emails under a business entity's domain name and instead tend to use public email services such as Mail.ru, Gmail, and Yahoo. Credible journals often register separate email addresses for each editor by subject area. Springer provides a list of editors for each department with their official e-mail (ch**@springernature.com), contact (+49-6221-487 ****), country (Germany), photos, and executive level (executive editor). On the other hand, Premier Publishing does not disclose its editors' contact information. Second, Springer lists 30 editors only for the physics department, whereas *European Science Review*, which Premier Publishing publishes, lists only 50 editors for 30 academic disciplines—this implies that only two editors are responsible for one academic discipline. Third, the editorial board of the physics department of Springer is geographically diverse and has editors from Germany, South Korea, China, the UK, the USA, and Japan—a sharp contrast to *European Science Review*, whose editorial team comes from only post-Soviet countries such as Kazakhstan, Russia, Uzbekistan, Kyrgyzstan, Ukraine, Azerbaijan, and Georgia, as well from India despite claims that the journal is "European". Premier Publishing is classified as a predatory publisher based on the editorial criteria.

### 5.4. Submission and Review Criteria

Most predatory journals have an undeveloped manuscript submission system. Consequently, they require authors to send their articles via e-mail or upload them directly on the website; however, credible journals usually develop their own manuscript submission system or use a third-party submission service, such as Manuscript Central—an online system used by journal editorial offices to manage and monitor the submission and peer-review process for articles, which is developed by the Scholar One platform. Springer uses the Editorial Manager manuscript-tracking system, where authors can monitor the publication status of their manuscripts. On the other hand, PP requires authors to upload their papers to the website directly. Further, journals published by Springer are usually peer-reviewed, which takes from at least one month to two years to complete. PP, in contrast, announces if the article is accepted or rejected within seven days, which is very suspicious given the limited number of editors. Certain fields such as physics or astronomy may require supplementary files such as astronomical images to ensure robust data; therefore, the publishers instruct authors on uploading their data, whereas predatory publishers lack such characteristics. The journal *Living Reviews in Solar Physics*, published by Springer, offers and provides instructions for authors under the "Electronic Supplementary Material" section. Premier Publishing, however, does not offer such features, although the journal publishes articles on physics; therefore, the submission and review criteria of *European Science Review* are not adequate to conclude that the journal is not predatory.

### 5.5. Ethics and Publication Fees

An article-processing fee (APF) is the key element determining predatory publishers. Sometimes known as a publication fee, the APF is a fee charged to authors to make their manuscripts available in open-access journals. These fees are necessary to cover publishing costs, including editorial costs and administering peer-review platforms. Predatory journals often charge relatively low prices to attract authors. In this regard, *European Science Review* charges USD 81 to publish an article up to six pages with an additional 20% for photos, diagrams, formulas, or charts in the article. Journals published by Springer charge different prices based on the subject, usually ranging from USD 1000 to USD 3000. It is worth mentioning that the article-processing fee is different from the publication fee. Credible publishers charge the former to provide several services to authors and make



their manuscript publicly available at no cost to readers. In contrast, predatory publishers charge the latter to publish articles on the website without providing a peer-review process.

Secondly, most editors of credible journals are members of the Committee on Publication Ethics (COPE). It is a not-for-profit organization that deals with publication misconduct, such as plagiarism, fraudulent and deceptive data, unethical research practices, duplication of works, leaks of confidentiality, and other research-related issues. Membership in COPE provides guidance and support to journal editors in dealing with unethical research practices. Most editors of Springer are members of COPE, whereas none of the *European Science Review* (ESR) editors are members that can be easily verified on the organization's website. The fact that ESR does not comply with ethical research practices can easily be verified by observing their journal issue of 11-12-2/2018 [28], on pages 152–153, where the confidentiality of individuals used as a subject for an experiment was not maintained. This is a serious breach of ethical conduct practices, given that the article was written by five authors (all of whom are from Uzbekistan), successfully went through a peer-review process, and was finally approved by the chief editor. Hence, the publisher fails to meet ethical and APF standards and does not qualify to be a credible publisher.

### 5.6. Website Design and Scope of the Journal

First, in many cases, predatory publishers display unrecognized and non-reputable indexing impact factors, such as the global impact factor (GIF) or the journal impact factor (JIF), to deceive authors. These misleading companies charge predatory journals for inclusion in the list. This method is adopted by Premier Publishing, whose main page indexes 10 misleading impact factors. On the other hand, Springer is indexed by official indexing systems such as Scopus and EBSCO. Second, it is typical for predatory publishers to have journals that are broad in scope. *European Science Review* is overly broad in scope because it does not specify its area of focus and publishes articles in 30 subjects ranging from economics to medicine in a single volume. In contrast, most reputable publishers choose specific areas of subjects such as *Living Reviews in Solar Physics*. Lastly, predatory journals run aggressive advertising campaigns by sending spam emails or calling for papers. Premier Publishing encourages authors to submit their work by distributing online advertising materials such as booklets, brochures, and catalogs.

Some of the predatory journals do not make the published journal issues available through their websites but share a private link to individual articles with the authors. The journal website provides neither the list nor the content of the entire issues and volumes. Some of these journals find it necessary to delete the content shared through private links, eliminating all footprints of the publication after a couple of months. For such journals, the materials of the most recent issues can be found only through search engines for a limited time. Articles published in such journals vanish and cannot be indexed or tracked further unless authors who have published these articles upload them to third-party platforms such as ResearchGate, Munich Personal RePEc Archive, or alike.

### 5.7. Qualitative Analysis by Universities

Figure 9 below compares universities based on the number of articles published in predatory journals. An overall picture emerging from this finding is that local universities and institutions are more likely to publish their articles in predatory journals. The National University, which ranks high among 125 institutions in terms of the number of publications, also ranks first based on the number of articles published in predatory journals. Tashkent University of Information Technologies has the highest rate of predatory publishing whose academics did not publish any articles in internationally recognized universities. This is closely followed by Tashkent Medical Academy, which published only four articles in credible international journals out of 122.

| | | Predatory articles | Credible articles | Predatory rate | Total articles |
|---|---|---|---|---|---|
| 📖 | National University | 167 | 52 | 76% | 219 |
| ➕ | Medical Academy | 122 | 4 | 97% | 126 |
| 🖥 | University of IT | 85 | 0 | 100% | 85 |
| ⚙ | Technical University | 79 | 43 | 65% | 122 |
| ➕ | Endocrinology Center | 63 | 10 | 86% | 73 |

**Figure 9.** Top-five universities based on the number of predatory articles.

Academics from foreign-accredited universities are less likely to publish in predatory journals. For example, from January 2016 to March 2019, only two academics from Turin Polytechnic University obtained D.Sc. and Ph.D. degrees compared to 81 academics from the National University of Uzbekistan. Two academics from the former university published 12 articles, of which 10 publications were issued in influential international journals such as *Optical Materials* (the Netherlands) and *Journal of the American Ceramic Society* (USA). Meanwhile, 81 academics from the National University published 219 articles of which 52 publications were issued in internationally recognized journals—this corresponds to 24% credible publications for the National University and 83% for Turin University.

This trend is possibly caused by a difference in professional requirements set by universities for career advancement opportunities. As Figure 4 demonstrates, only academics from state universities actively publish their research; thus, analytical comparison between universities demonstrates that there is a strong incentive for the scholars of state universities and institutions to obtain Ph.D. and D.Sc. degrees compared to foreign-accredited universities in Uzbekistan.

Figure 10 below provides a comparison of top-five universities that actively publish articles in reputable journals.

| | | Credible articles | Predatory articles | Credibility rate | Total articles |
|---|---|---|---|---|---|
| 📖 | Samarkand State Uni. | 54 | 42 | 56% | 96 |
| 📖 | National University | 52 | 167 | 24% | 219 |
| π | Inst. of Mathematics | 46 | 18 | 72% | 64 |
| ⚙ | Technical University | 41 | 81 | 34% | 122 |
| 🧪 | Inst. of Chemistry of Plant Substances | 37 | 5 | 88% | 42 |

**Figure 10.** Top-five universities based on the number of scholarly articles.

Samarkand State University, with a total of 54 publications, comes first, with most articles being published in the field of physics (47%), philology (18%), chemistry (9%), and astronomy (8%). The National University of Uzbekistan, which ranks first in terms of the number of articles published in predatory journals, comes next, whose academics published 52 articles in internationally reputable journals. Similarly, Technical University, the fourth-highest-ranked university based on the number of predatory publications, is in the fourth place based on credible publications. The Institute of Mathematics and the Institute of Chemistry of Plant Substances are also included in the top-five list. Despite publishing fewer articles cumulatively, the quality of the journals in which their researchers publish is high, at 72% and 88%, respectively.

Figure 11 below demonstrates the top-five academic disciplines in which Uzbekistani scholars publish credible articles. The top subjects are led by physics and chemistry,

each of which accounts for 41% and 17% of the total publications in globally recognized journals, respectively. Although not as productive as the other disciplines in terms of quantity, physicists and chemists have achieved several impressive and globally well-recognized results.

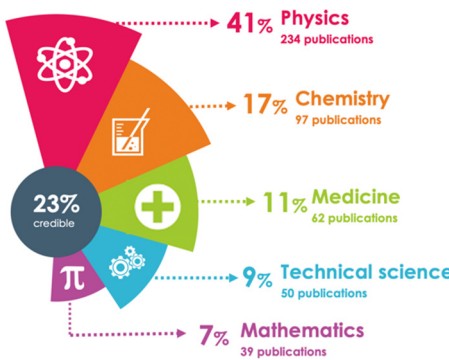

**Figure 11.** Top-five academic disciplines based on the number of credible publications.

As Abdurakhmanov [28] wrote: "For the first time in the world, stable superconducting materials with increased transition temperature to the superconducting state up to 110–150 degrees Kelvin have been obtained with the help of concentrated solar energy". In chemistry, Uzbek scientists developed a new research field—adsorption-energy stoichiometry—with new concepts, theories, and semiempirical results published in an article by Springer [29]. Uzbekistan led the way on this subject because several scientific research institutes and centers in Uzbekistan focus on the relevant fields, such as the Institute of Nuclear Physics and the Institute of the Chemistry of Plant Substances; however, this is not the case for all academic disciplines.

Figure 12 illustrates the top-five fields where academics are mostly involved in predatory publishing practices. The data show that researchers from the medical field are more likely to publish articles in predatory journals.

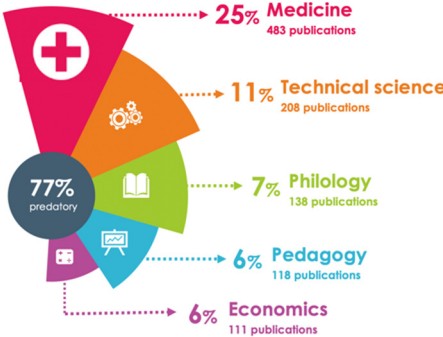

**Figure 12.** Top-five academic disciplines based on the number of predatory publications.

Those in the medical field fall prey to predatory journals because obtaining Ph.D. and D.Sc. degrees in medicine is necessary for promotion to higher positions and degrees and, correspondingly, a requirement for increasing their wages. The next four subjects in which researchers are more likely to submit their papers to predatory publishers are technical sciences, philology, pedagogy, and economics. The main reasons behind this trend are (i) the lack of dedicated research institutions in those fields—the articles are published by university lecturers for the sake of publication but not as a result of conducting original research; (ii) reputation pressure to publish in international journals, especially by universities, where the payoff for real research is minimal, which further stimulates predatory publishing; (iii) the lack of thematic research funding agencies or departments

within the organization with research support tools and resources; and (iv) a lack of competence for distinguishing credible publications from predatory publications.

### 5.8. Qualitative Analysis by Country of Publication

Nearly a quarter of all articles issued in scholarly journals are published in Russia. In practical terms, this corresponds to 137 publications, of which 55% were published by physicians in journals such as *Semiconductors, Optics, and Spectroscopy* and *Theoretical and Mathematical Physics*. Following Russia, 93 articles were published in influential journals in the USA. Academics in astronomy (33%) are more likely to publish in scholarly journals in the USA, such as *Astrophysics*. In the Netherlands, 79 papers were published; the top fields include physics, mathematics, and medicine; the journals include *European Journal of Heart Failure*, *Linear Algebra and its Applications*, and *Astrophysics and Space Science*. Lastly, in the UK, 19 articles in chemistry, 16 articles in medicine, and 8 articles in biology were published by Uzbek scientists.

Uzbekistan is not the only country where publish-or-perish policies are implemented. Most post-Soviet countries such as Kazakhstan and Ukraine have similar academic requirements for Ph.D. and D.Sc. candidates. Most of those countries have similar economic climates, and researchers are expected to handle the financial burden of their publications. This, in turn, increased the demand for low-budget online publications, creating a business opportunity for predatory publishers to flourish.

Unsurprisingly, most predatory publishers are registered in European countries and the USA because researchers believe that publishers based in these countries are more credible and reliable. Researchers from Uzbekistan are more likely to publish in predatory journals registered in Austria and Germany primarily because those publishers name their journals by including buzzwords, e.g., "European". Similarly, predatory publishers from India usually name their journals by including the keyword "international" to seem prestigious. For instance, *International Journal of Advanced Research* and *International Journal of Science and Research* are registered in India. Similarly, to deceive authors, predatory journals registered in the USA usually imitate the names of scholarly journals published in the USA. For instance, *The American Journal of the Medical Sciences* is changed to *American Journal of Medicine and Medical Sciences*. Thus, our findings suggest that most journals containing one of those buzzwords are highly likely to be predatory.

### 5.9. Regression Results

The results presented in Table 4 provide insight into what factors shape the trustworthiness of peer-reviewed journals. The results have the interpretation of ceteris paribus—other things equal. The model suggests that if the journal's name contains one of the suspicious keywords, such as "scientific" and "research", the probability of the journal being predatory increases by approximately 35%. The fact that the journal claims to be international tends to further drive the untrustworthiness of the journal by 20%. For instance, since the journal *European Science Review* contains both keywords "European" and "science", the probability that this journal is predatory increases by 55%. Concerning the date of publication, the probability of articles being published in predatory journals decreases by nearly 37 percent if a publication was issued before 2011.

This result agrees with the proposed hypothesis, which suggests that, in response to the open-access model, which shifted the financial burden from readers to authors, academics in Uzbekistan are now more likely to publish in predatory journals due mainly to financial constraints.

The probability of an article being published in predatory journals decreases by nearly 4.5% if a researcher aims to defend a Doctor of Science degree. The number of words in the journal's name also has significant explanatory power. As the number of words increases by one, the probability that the journal is predatory increases by two percent. For example, *International Journal of Innovative Science, Engineering, and Technology* contains six words, which means that there is a 12-percent probability that this journal is predatory. If other

factors are also considered, such as the two keywords "international" and "science", the estimated probability of the journal being predatory is 67%.

**Table 4.** Regression results.

| | LPM | Logit | Probit |
|---|---|---|---|
| International | 0.1946164 | 2.15053 | 1.216796 |
| | (0.000) | (0.000) | (0.000) |
| Scope | 0.3488624 | 2.765282 | 1.57292 |
| | (0.000) | (0.000) | (0.000) |
| Word count | 0.0210394 | 0.1745708 | 0.1067014 |
| | (0.000) | (0.000) | (0.000) |
| Degree | −0.044665 | −0.7152397 | −0.4025087 |
| | (0.000) | (0.000) | (0.000) |
| Before 2011 | −0.3663505 | −2.027902 | −1.099638 |
| | (0.000) | (0.000) | (0.000) |
| Constant | 0.4072716 | −0.7624518 | −0.4729806 |
| | (0.000) | (0.000) | (0.000) |
| $R^2$/Pseudo $R^2$ | 0.4918 | 0.5019 | 0.5079 |
| Adjusted $R^2$ | 0.4908 | | |
| Observations | 2502 | 2502 | 2502 |

Unlike the LPM, the slope coefficients do not provide the rate of change of probability for a unit change in regressors. For the logit model, a more meaningful interpretation is in terms of the odds ratio obtained by taking the antilog of slope coefficients. Regarding the first variable, a journal that claims to be internationally recognized is more than 8.59 times as likely to be predatory than those that do not, ceteris paribus.

Journals whose titles include such keywords as "scientific" or "research" are more than 16 times as likely to be predatory. As the number of words in the title of the journal increases by one word, it is expected to be more than 1.2 times as likely to be predatory; however, the interpretation of coefficients in the logit model when they are negative differs from the LPM. For instance, concerning an academic degree, only half of the candidates defending D.Sc. titles are likely to publish in predatory journals compared to Ph.D. levels. Only one-tenth of the articles published before 2011 are likely to be issued in predatory journals. Qualitatively, the results of the probit model are comparable to those obtained from the logit model. In short, journals claiming to be international and broad in scope and the number of words in a journal's title are likely to drive the probability of the journal being predatory, whereas the other two variables, a degree of academic level and journals published before 2011, are negatively related to the probability of the journals being predatory.

## 6. Conclusions

The root cause of the problem originates from the imposition of annual publication requirements unsustainable for most researchers by the Supreme Attestation Commission of the Republic of Uzbekistan for Ph.D. and D.Sc. candidates. Consequently, local universities adopt these requirements as a benchmark for promotion and bonuses. In turn, this drives lecturers who want to stay in academia to engage in pseudo-research.

In western countries, an article-processing fee (APF) is usually financed by an author's institution or through dedicated research funds. In Uzbekistan, however, authors have to bear the financial burden themselves. The monthly income of researchers in Uzbekistan is too low to afford the APF of decent open-access journals; therefore, most researchers submit their manuscripts to predatory journals, which charge as little as USD 80 for a publication. The impact of this can be seen in Figure 13, where the percentage of articles published in scholarly journals was decomposed into two periods, before and after the open-access publishing policy was implemented.

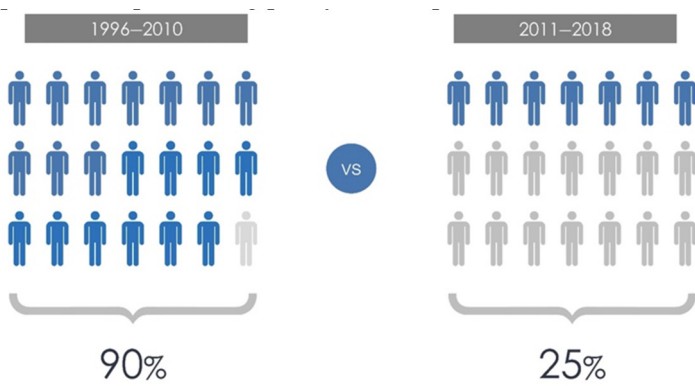

**Figure 13.** Percentage of articles published in credible journals before and after the introduction of the open-access model in Uzbekistan.

Although 9 out of 10 researchers published their research in internationally reputable journals, only one out of four authors submit their publications to scholarly journals. The larger part of the 25% of credible publications is published primarily by physicists and chemists who publish their works in the *Applied Solar Energy* and *Chemistry of Natural Compounds* journals currently operated by Springer and originally established in Uzbekistan. Along with being subscription-based journals operated in three languages, these journals are affordable and accessible to local scholars.

None of the universities in the country operate a rigorous and well-designed research facilitation policy. The Supreme Attestation Commission's international publication requirements created practically unachievable objectives for university lecturers. High-quality research can only be produced by allocating sufficient time and funds and enabling a decent level of hard and soft research infrastructure and environment. Despite research institutes meeting these conditions in part or in full, none of the universities, even the most prestigious, operate such a research facilitation policy. The most worrying aspect of the currently operated "publish or perish" policy is the expected publication frequency. Some universities require one publication a year; some universities require two publications in an indexed journal per annum—this results in university lecturers being obliged to publish with predatory publishers voluntarily.

Lack of funds dedicated to cover the publication-related costs is another reason for increasing trends of predatory publishing. Researchers who can produce quality research appropriate for reputable journals cannot cover the APF of the credible open-access journals—their only option is to publish in low-cost predatory journals.

Our findings suggest that the expectation that researchers publish in international publications has shifted the focus of research from conducting high-quality research that contributes to science or society to publishing research to maintain their academic position, remain competitive with their peers, and secure promotions. In turn, this has led to unethical practices and waste of research resources. The responsible authorities, the Supreme Attestation Commission and university administrations being the most important players, have set unattainable targets that harm research culture and environment.

The main purpose of research is to ensure that the community or industry benefits from its results. Such research is usually time-consuming. In general, the period of time between the first and second publication is usually expected to be longer; however, most of the candidates who obtained an academic degree between 2015–2019 produced at least two to three publications in international journals in one year—this implies that a candidate conceived an idea, submitted a protocol, received institutional approval, undertook the research, wrote the paper, went through peer-review successfully, and finally published it within six months. It is practically impossible to go through these processes within such a short period given that most applicants work as teachers at universities. Since the present study only deals with international publications, the articles published in national journals were not included for the purposes discussed in the methodology section. In that case, it

would correspond to 30–60 articles per author. This practice shows that, currently, only predatory journals benefit from the "publish or perish" policy; researchers and university lecturers are suffering along with the country's reputation in scientific research.

Institutes and universities should expect lecturers and scholars to regularly publish in indexed, credible, and reputable journals and make funds available for publication expenses. This fund could be spent on covering the journal APF and for additional quality assurance services, such as language-editing services.

The above analysis revealed that university lecturers are the main clients of predatory publishers, followed by scholars attempting to earn degrees. Universities should develop and adopt pragmatic, attainable, and graded requirements according to the lecturers' grades with transparent key performance indicators, which, instead of immediate and frequent publication requirements, promotes the eventual growth of research capacity.

International universities operating in Uzbekistan could play a leading role by developing and implementing a well-designed research support policy. Despite the significant allocations to research and development in their budget, foreign universities have insignificant contributions to "credible" publications and do not increase the country's reputation. None of these universities are known to cover the APF of their staff once their article is accepted. Moreover, while national universities revise their workload policies, intending to allocate more time for research, international universities operating in Uzbekistan have been increasing the teaching hours of the staff—these universities should revise their inefficient research support policies.

Conducting research and publishing an article are two separate time- and resource-consuming processes; therefore, authorities should modify publication requirements by providing more time to publish in credible journals. They should also replace the quantitative requirements, such as "one article in an international journal per year for a professor", with a dynamic qualitative indicator, such as "three-year rolling average *h-index*" or "three-year rolling average impact factor". The funds being allocated from the state budget for funding research should be spent covering the APF of the authors once they are accepted to OA journals. The policy of offering national standard D.Sc. degrees to Ph.D. degree holders from the top 1000 universities should also be enriched by evaluating the third-party-proven publication history of the degree holders.

Most importantly, the Supreme Attestation Commission should terminate the recommended list of journals for publication. The local web portals used for evaluating and ranking the scholars, such as www.salohiyat.uz and www.fan-portal.mininnovation.uz, have to be integrated into recognized databases, such as the Scopus Authors database, SSRN, ORCID, ResearchGate, Google Scholar, and RePEc/Ideas platforms.

**Author Contributions:** Conceptualization, B.E. and R.E.; methodology, K.A. and B.E.; formal analysis, K.A. and B.E.; resources, K.A. and B.E.; data curation, K.A.; writing—original draft preparation, K.A. and B.E.; writing—review and editing, M.I. and R.E.; supervision, R.E. and M.I. All authors have read and agreed to the published version of the manuscript.

**Funding:** This research received no external funding.

**Data Availability Statement:** The data presented in this study are available on request from the corresponding author.

**Acknowledgments:** The authors are boundlessly thankful to the three anonymous referees for their reviews, feedback, and suggestions, which have taken this manuscript to another level.

**Conflicts of Interest:** The authors declare no conflict of interest.

## Notes

[1]     Excerpt of item 17 from the Cabinet of Ministers Resolution No.304 from 22nd of May 2017, main regulatory document on obtaining scientific degrees.

[2]     Education portal of Uzbekistan—ZiyoNET: http://library.ziyonet.uz/ru/book/index (accessed on 29 July 2021).

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
