# Peer review of "Efficiency of “Publish or Perish” Policy—Some Considerations Based on the Uzbekistan Experience"

_publications, doi:10.3390/publications9030033_

Round 1

Reviewer 1 Report

The paper states that a majority of articles written by authors from Uzbekistan are published in predatory journals. The definition of predatory journals is based on the work by Jeffrey Beall (Beall's list). However, Beall's list has been criticised - which could have been verified with a simple Wikipedia search.

There are other initiatives to help auhors finding reputable journals, such as the Think.Check.Submit website (https://thinkchecksubmit.org/about-2/).

Furthermore, the authors show little understanding of open access publishing. The first assumption is that open access publishing is synonym with the author-pays model. A quick search in the Directory of Open Access Journals (https://doaj.org) reveals over 11,000 journals that do not charge fees to authors - https://bit.ly/362S1LJ.  

The second assumption seems to be that open access publishing is of lower quality. Again, this needs to be nuanced. For instance, the DOAJ has removed over 3,000 journals to ensure that quality criteria are met - see also http://dx.doi.org/10.4403/jlis.it-12052. 

I understand the issue the authors are raising: forcing authors to publish often in 'international' journals might have an adverse effect. However, to prove this statement, the authors must present a better definition of what type of journal is below the required standard. This definition should be grounded on more than one disputed source.

Author Response

Dear Reviewer,

We are thankful to all reviewers for their precious time and efforts spent for reviewing our manuscript. We are especially indebted for the insightful reflections, comments and suggestions.

And we are very sorry to create an impression that open-access publishing has led to negative consequences. We want to highlight the fact that “publish or perish” policy implemented in Uzbekistan barehandedly without linking it to valid and tangible Key Performance Indicators can have a rebound effect and discredit the country’s reputation. And we have emphasized that possibility of open-access publishing was used as a tool to work-around the extremely high requirements set by the Supreme Attestation Commission.

Therefore, principal contribution of this study is qualitative and quantitative analysis of the publish or perish policy in Uzbekistan.

We have extensively revised the manuscript with an overarching aim of:

  • Providing objective information about the need for open-access publishing, why it emerged and why it is necessary in the current context;
  • What particular problems are solved due to emergence of open-access publishing;
  • What are the obstacles for scholars from the low and lower-medium income countries for publishing their works in open-access form;

We have reviewed the policy implications section of our study accordingly. Please find our responses below

Once again many thanks for your comments and suggestions.

Best regards,

Bahtiyor

Reviewer 2 Report

I am not quite sure what the main object of this article is. Part of it seems to be directed at discussing the problems of PhD and DSc students in Uzbekistan. Part seems to be aimed at underlining the deleterious effects of predatory publishing. The relationship between these two is never quite clear, and it might be better to treat them separately in two different articles. If not, it will be necessary to show how they are related and why it is necessary to treat them together. As a result the whole of the paper lacks clarity, and the argumentation is obscure.

Some of the Figures are not clear. What do the three numbers in Figure 1 represent? Does Figure 2 mean that there were no students who obtained a PhD in 2016? The legend to Figure 5 says that it indicates the number of articles per author, but the histograms appear to give numbers of authors, not number of articles.

No distinction is made between science and humanities. It is well known that while the percentage of articles in the scientific sector is over 90%, the figure is much less in the humanities. This must affect the figures in some way, and should therefore be taken into account.

In 5.7, the first and second paragraphs are almost identical (perhaps the second is a re-writing of the first, and the authors forgot to delete the first).

Overall, I find that the paper conflates a number of questions that would be better dealt with separately. First the question of the difficulties of PhD and DSc students, and secondly the problems raised by predatory publishing. Even this latter question conflates the question of the dangers of predatory publishing with that of recognizing predatory publishing. Moreover, the writing is fairly repetitious, and frequently doubles back on itself. As a result the paper as a whole lacks clarity.

In addition, the English in which this paper is written is below the standard required for publication in an international journal. In some sections, the problem is much more acute than in others, as though different members of the team, with differing levels of competence in English, had drafted different parts of the article. The use of the English system of determiners seems to pose a particular problem. I am all too aware of the extra difficulties this requirement places on non-anglophone writers, but they should not underestimate the importance of correct English for international publication.

Consequently, the article cannot be published as it stands. I recommend that the authors write up their material as two (or even three) separate articles, paying particular attention to clarity and sound argument. They should then have their texts vetted by an anglophone who is familiar with this field. Even so the resulting articles will have limited interest, since the research relates only to Uzbekistan, and will therefore be mainly of interest to researchers in that and neighbouring countries. It might therefore be more appropriate to seek publication in a more local publication than an international one.

Author Response

Dear Reviewer,

We are thankful to all reviewers for their precious time and efforts spent for reviewing our manuscript. We are especially indebted for the insightful reflections, comments and suggestions.

And we are very sorry to create an impression that open-access publishing has led to negative consequences. We want to highlight the fact that “publish or perish” policy implemented in Uzbekistan barehanded without linking it to valid and tangible Key Performance Indicators can have a rebound effect and discredit the country’s reputation. And we have emphasized that possibility of open-access publishing was used as a tool to work-around the extremely high requirements set by the Supreme Attestation Commission.

Therefore, principal contribution of this study is qualitative and quantitative analysis of the publish or perish policy in Uzbekistan.

We have extensively revised the manuscript with an overarching aim of:

  • Providing objective information about the need for open-access publishing, why it emerged and why it is necessary in the current context;
  • What particular problems are solved due to emergence of open-access publishing;
  • What are the obstacles for scholars from the low and lower-medium income countries for publishing their works in open-access form;

We have reviewed the policy implications section of our study accordingly. Please find our responses below

Once again many thanks for your comments and suggestions.

Best regards,

Bahtiyor

Reviewer 3 Report

The manuscript of Eshchanov and coworkers describes the effect of the Publish and Perish mandate in a country where academics are faced with high demand and low support. They are expected to publish two or more papers a year, but without funds to cover the publication fee / article processing charge they turn to cheaper publishers who do not offer the level of professionalism (euphemism for being predatory) as more established publishers.

I think the topic is of great importance. While the predatory journals pop up with European or US head offices to increase credibility they cater to scientist from other regions. I recommend major revision, not because I do not feel the manuscript is worthy (it is), but because with a restructuring and some clarification it could be much better.

I think the paper needs a restructuring. In the beginning it seemed to me that the authors want to imply that asking for publications to award a PhD or a DSc is somehow bad. If so please explain why! It could be that the threshold is so high that very few could pass, especially without resorting to publication in dubious journals. In the discussion, however, the authors state the high demand on yearly publication as the major problem. The two, asking for a number of scientific papers to award a PhD or DSc, or demanding 2+ publications per year without adequate support and without allowing for a moving window is two very different beast. The end-results of scientific inquiry are scientific papers. Without them the results are not available to the scientific community and thus do not enrichen humanity’s collective knowledge. So there are usually qualification thresholds for PhD, habilitation and DSc.

If I may suggest, the manuscript should start with a more in-depth description of what are the requirement for a PhD or a DSc is in Uzbekistan. There are regional differences in what are the requirements for a PhD. Just for comparison, in Hungary in biology at Eötvös University (because it can vary from University to University, and from discipline to discipline) a PhD can be awarded for two papers, one of which needs to be a first author paper by the candidate. PhD became an entry level requirement in academia. Position for adjunct professors (assistant professors) or research fellows requires a PhD, and pretty much any research grants also requires one. PhDs are awarded by the Universities. But DSc is awarded by the Hungarian Academy of Sciences, it is a de facto requirement for full professorship and has some criteria on number of papers, citations, grants, finished PhD students, memberships in organisations, etc. Please note that at least I know what a DSc is, as Hungary, being a post-Soviet bloc member, has a similar academic system as post-Soviet Union states. DSc is not part of the main-stream Anglo-Saxon academic advancement; thus it needs an explanation for the international readers. Consequently, greatly expand the description in lines 201–207.

The authors themselves cite that the reason behind resorting to predatory journals might not be the cost factor, but the poor quality of the paper. The main feature of predatory journals is not the low cost (OA in certain fields and at journals operated by societies and universities or heavily subsidized by other sources might not ask for any fee and still make papers publicly available), but the lack of peer-review. It is definitely easier to have papers if there is no quality or plagiarism checks in place. There is actually a term, vanity publication, which covers publication just to bolster one’s list of papers.

What do the authors think, which is more important: the lack of adequate funds or the poor quality of papers that pushes scientist in their country toward predatory journals? I’m asking this, as the financial situation is often grim here as well, but then we just turn to subscription-based journals to publish our work in. They are in this sense free. We have a mandate (a strong suggestion) to publish our papers in open access journals or at least make them available to the public, but this can be done by posting the last, not edited version in a national publication archive. So why not just go for the non-OA journals when funds are low in Uzbekistan?

There are very few citations in the manuscript. The topic of publish and perish has some literature, there might be papers on the transformation of academia in the eastern bloc as well.

Minor comments

Avoid the use of [ibid], I know it is frequently used in the humanities, but not in the natural sciences. There is no problem having the same reference after one another.

Figure 1. Display the years (2016-2017-2018-2019) on the figure. It is only available in the main text, and a figure with its caption should be self-contained and understandable.

In figure 3 and onward: If you know that “technical sciences” are the right term, then it is fine. I have the vague feeling that it covers engineering (electrical-, mechanical-, chemical-, civil engineering, architects, etc.).

Table 1. This data does not require a table, just write them into the text.

Table 2. At the ethic/publication fee line, it is about membership in COPE not just is/not COPE.

P9L372: Netherland -> the Netherland

Section 5.5: Please be consistent with article processing charge or fee. They are the same, just read better if it is consistent.

P1L507 The sentence starting with “Along with” seems to be unfinished and strange.

Figure 12. Does it mean that out of 77% of predatory publications 25% comes from medicine, 11% from engineering, etc.? Please elaborate in the caption.

P14L644-649: The authors meticulously list where certain journals come from where their fellow countrymen publish (P14 top). But at the same time, they write earlier, that predatory journals often have headquarters in the USA, Austria, etc. So does the country of origin matters for a journal?

P14L651-656: If you know, please elaborate on the system in Kazakhstan and Ukraine.

P14L681-690 it is a repetition of the previous paragraph.

P17L800: Consider “it would correspond to from 30 to 60”

P17L806: Consider “should not be limited to”

Author Response

Dear Reviewer,

We are thankful to all reviewers for their precious time and efforts spent for reviewing our manuscript. We are especially indebted for the insightful reflections, comments and suggestions.

And we are very sorry to create an impression that open-access publishing has led to negative consequences. We want to highlight the fact that “publish or perish” policy implemented in Uzbekistan barehandedly without linking it to valid and tangible Key Performance Indicators can have a rebound effect and discredit the country’s reputation. And we have emphasized that possibility of open-access publishing was used as a tool to work-around the extremely high requirements set by the Supreme Attestation Commission.

Therefore, principal contribution of this study is qualitative and quantitative analysis of the publish or perish policy in Uzbekistan.

We have extensively revised the manuscript with an overarching aim of:

  • Providing objective information about the need for open-access publishing, why it emerged and why it is necessary in the current context;
  • What particular problems are solved due to emergence of open-access publishing;
  • What are the obstacles for scholars from the low and lower-medium income countries for publishing their works in open-access form;

We have reviewed the policy implications section of our study accordingly. Please find our responses below

Once again many thanks for your insightful comments and suggestions. They were exceptionally useful!

Best regards,

Bahtiyor

Reviewer 4 Report

I stand by my previous comments regarding the introduction, which were neither "subjective" nor "prejudice" as the authors' response stated. I did not say that the study had not been conducted recently, I said that the introduction relied "heavily on references published in the period 2002-2007" and that this gave the impression that the introduction had been written some time ago and not updated with more recent research.

For example, in lines 77-79 the authors state: "[the] libraries of the University of California and Cornell capitalized USD 8 million and 1.7 million, respectively, to Elsevier in 2003", a statement that the authors support with references published in the period 2002-2004. Since the University of California's dispute with Elsevier has been headline news recently, there are plenty of more up-to-date estimates of how much this particular university spends on Elsevier subscriptions that could have been quoted instead. In this news article we see that in 2019 when the dispute reached an impasse, UC was paying $11 million per year to Elsevier. This article from March this year regarding the resolution of the aforementioned dispute states that "California and its scientists will keep paying Elsevier about $13 million a year" for a read-and-publish deal. My point is that the value quoted by the authors for this particular deal is almost twenty years out-of-date, and is based on outdated sources when many more recent estimates (and sources) are easily available.

The same holds for their estimates for rises in subscription fees (lines 74-75), which are based on data collected in the period 2001-2005 (reference #4) and a paper from 2007 which I couldn't access because the url given in reference #5 seems to be incorrect. There are far more recent sources of data on rises in subscription costs that could have been added. For example, the data referred to in this report, which covers the period 2010-2014 and are available here. These show that for the period 2010-2014, average subscription costs increased - but were flat (in fact showing a slight decline) when corrected for inflation.

It is for these reasons that I am surprised that the authors can "declare with confidence that [they] have included all the relevant studies in the current version of the paper." They have not, and I hope that I have satisfied the authors' demand that I "point-out any relevant study which is omitted from the literature review." I have now done so.

The argument presented in lines 100-106 (supported by a single reference to Jeffery Beall's article) that subscription price rises can be explained by baby-boomers completing their PhDs combined with "the advent of new fields" neglects a host of other factors such as the commercialization of scientific publishing (that began while many baby-boomers were still babies), and the fact that the cost of journals is largely hidden from the "consumer" which alters the usual supply-demand relationship. It also ignores the fact that "new fields" have been proliferating since the very dawn of science, but for a long time most journals were loss-making, not profit-machines. In short, Beall's argument doesn't stack up and so it should not be given the prominence that it is here.

What is most problematic is the disconnect between the introduction and the study itself. I have some small qualms about the study itself, but they are relatively insignificant compared to the fact that there is a lot of material in the introduction that is simply not relevant to the study, and is sometimes controversial, or supported by outdated sources as I have tried to demonstrate. It is simply not necessary for the authors to review the entire history of Open Access, discuss the costs of subscriptions, and the serials crisis, etc. The authors should focus on the phenomenon of predatory publishing (and perhaps survey the literature for opinions that differ from, or are more nuanced than those of Jeffery Beall) and the situation regarding perverse incentives in Uzbekistan - which sounds fascinating, but which I struggled to understand fully from the introduction. In short, the paper needs to be much more focused.

Minor issues:

My point regarding the rate of “predatory” publishing simply being the inverse of the rate of “credible” publishing still stands. Just pick one of these to present and show the top 5, and the bottom 5 according to this metric.

I couldn't access the text of references #5 and #11 because the urls given seem to be incorrect. Please update these.

Reference #12 does not seem to be referred to anywhere in the main text.

The acronym CPI (line 75) is not defined, and the acronym APF (line 92) needs to be defined at first use.

"Frontiers" is spelt wrong on line 159

Author Response

Dear Reviewer,

We have an impression that you don't look at the broad picture. Instead, you pick up individual statements/arguments out of context. Then you construct your arguments around the individual statements that are fragmented from the context.

We have provided the information on subscription fees to demonstrate that subscription fees are high and constantly growing, making them unaffordable for institutions of third-world countries. The most recent incident and the deal between the University of California and Elsevier is not the topic of our discussion. But previous studies provide the University of California and Iowa cases as an example for high subscription fees, which eventually was used as one of the arguments for the emergence of open-access publishing.

It has become clear to us that you don't accept the framework developed by Jefferey Beall. We agree that it has some shortcomings, but we also find it a well-documented and broadly accepted framework with tangible indicators for evaluating the journals.

The introduction is amended to present the goals and objectives of the study immediately at the beginning. In addition, we have eliminated some of the inappropriate statements and irrelevant information from the introduction.
Please find our point-by-point response in the attachment.

Best regards,
Authors

Round 2

Reviewer 1 Report

Dear authors,

There is one issue that I think should be addressed. The introduction states that open access publishing is always based on a business model where the author pays for publication, which leads to the predatory practices you describe. This is not the case.

There are several other business models in open access journal publishing that do not rely on author funding. I would highly recommend to describe this in the introduction. A short overview and some useful references might be found here: https://en.wikipedia.org/wiki/Open_access

Author Response

Dear Reviewer,

Many thanks!

Bahtiyor

Reviewer 2 Report

This version is a slight improvement on the first, but the points I raised in my first review remain largely true here. This includes the language, which is far below the standard necessary for publication: I counted 10 errors in the Abstract alone, and the main text is of the same order, to the point of sometimes impeding understanding.

Since this is a second submission, I regret that in the circulstances I can only recommend rejection.

Author Response

Dear Reviewer,

Dear Editors,

I have made further small changes to the manuscript based on the suggestion of the other reviewer.

We are non-native English speakers. We understand that there may be many omissions and shortcomings related to the language. Therefore we agree to use the MDPI language editing services.

Please let us know if this is feasible at this stage.

Best regards,

Bahtiyor

on behalf of the authors

PS latest version of the manuscript attached here. I could not attach it elsewhere, apologies

Reviewer 3 Report

The revised manuscript of Eshchanov and coworkers has improved and mostly had taken my advice for restructuring. I like the conclusion. The repetition of the criteria of being a predatory journal and the regressions there is something that I would not put into this publication (but it is your paper and not mine). I very much liked the conclusion part.

As far as reference to prior literature is concerned, it still only deals with OA and predatory publishing, and not on the publish and perish phenomenon, or situation in other post-Soviet places. Just two papers (none of which are mine):

  • Kozak, M.; Bornmann, L.; Leydesdorff, L. How have the eastern European countries of the former warsaw pact developed since 1990? A bibliometric study. Scientometrics 2015, 102, 1101–1117.
  • Grančay, M.; Vveinhardt, J.; Šumilo, Ē. Publish or perish: How central and eastern European economists have dealt with the ever-increasing academic publishing requirements 2000–2015. Scientometrics 2017, 111, 1813–1837.

Just my thought on your ideas on how to change the system (so you don’t have to do anything with it, just read it): I sincerely think that scientist should not be forced to publish at any predetermined rate. I know that many would counter, that then people would not work at all, or not publish at all. This is probably not true, at least not for all scientists. As you have rightfully observe, university professors (at any level) spend considerable amount of time teaching (surprisingly to some in commitees…) and the pace they can proceed with their research is lower compared to ones working in scientific institutes without teaching obligations.

The rolling average of number of papers or anything is a must, as some papers can stay in review for 2 years as you have written (especially in some disciplines, like economy). Impact factor has its own problem, and attainable numbers vary wildly between disciplines. For example, 3 is already good in ecology, but not that great in medicine.

Furthermore, why should the Supreme Attestation Committee stop having a list of accepted OA journals? If they would only list reliable ones, then it is fine. Sometimes these lists have native journals that are seldom indexed in the big databases, but are still valuable and respected national journals, and publication in them might count toward promotion.

Minor comments

(Line numbers are for the track changed version, that I have received from the publisher)

L17 I would rephrase the sentence to something like this: “Thus, it leads to high pressure on academia to publish.”

L18 What is KPI in the abstract? Please try to avoid abbreviation in the abstract.

L57: There is a leftover comment to have the requirement for DSc/PhD here. DSc requirements are given later, but not PhD requirement. You should at least give it for your university/discipline.

L71 “Subscription-based publishing has been the conventional model”

L76 For instance, the University of Iowa spent

L90 This sentence does not read well with the modifications

L100 Omit “The” from the beginning of the sentence

L102 “to accommodate ever-increasing number of research papers”

L113 “subscribe to obtain access to key journals”

L135 “irrespective of its geographical location”

L137 “not only saves time but”

L199 “reputable international journals to obtain a degree or promotion”

L198-200 Please reread this sentence, I think this is what you wanted to write: “Especially, when young researchers are put under pressure to publish their work in international journals to obtain a degree or promotion, predatory journals flouris.”

L203 This paragraph seems out of place, or needs some beginning to fit with the other part of the section.

L244 I would not start a sentence with “E.g.”, consider replacing it with “For example,”

L301 “which collectively accounting for 50% of the total”

L380 “In contrast to Kazakhstan’s case”

L384 I think nor “below” nor “on the next page” is required. The final layout might change.

L471 The country “the Netherlands” is usually spelled like this, with a definitive article before it. Thus the sentence should be written as “… and are translated into English in Russia and the Netherlands where …”

L525 Consider “editors from Germany”

L636 Netherlands -> the Netherlands

L716 Netherlands -> the Netherlands

L880 “for the purposes discussed in the methodology section”

L887 “This fund could be spent on covering the journal APF”

L889 “The above analysis”

L891 Something is wrong here “develop and adopt pragmatic, attainable for lecturers, level-gradient research”

Ref 25: “Does it take too long to publish a research”

Author Response

Thank you for your comments. The paper is revised accordingly. Please see the attachment.
